# Economic Evaluation of Oral Cancer Screening Programs: Review of Outcomes and Study Designs

**DOI:** 10.3390/healthcare11081198

**Published:** 2023-04-21

**Authors:** Sivaraj Raman, Asrul Akmal Shafie, Bee Ying Tan, Mannil Thomas Abraham, Shim Chen Kiong, Sok Ching Cheong

**Affiliations:** 1Centre for Health Economics Research, Institute for Health Systems Research, National Institutes of Health, Shah Alam 40170, Malaysia; 2Institutional Planning and Strategic Center, Universiti Sains Malaysia, Penang 11800, Malaysia; aakmal@usm.my; 3Discipline of Social and Administrative Pharmacy, School of Pharmaceutical Sciences, Universiti Sains Malaysia, Penang 11800, Malaysia; 4Oral and Maxillofacial Surgery Department, Hospital Tengku Ampuan Rahimah, Ministry of Health, Klang 41200, Malaysia; 5Oral and Maxillofacial Surgery Department, Hospital Umum Sarawak, Ministry of Health, Kuching 93586, Malaysia; 6Digital Health Research Unit, Cancer Research Malaysia, Subang Jaya 47500, Malaysia; sokching.cheong@cancerresearch.my; 7Department of Oral and Maxillofacial Clinical Sciences, Faculty of Dentistry, University of Malaya, Kuala Lumpur 50603, Malaysia

**Keywords:** oral cancer, screening, cost-effectiveness, economic evaluations, study design, modeling

## Abstract

A lack of guidance on economic evaluations for oral cancer screening programs forms a challenge for policymakers and researchers to fill the knowledge gap on their cost-effectiveness. This systematic review thus aims to compare the outcomes and design of such evaluations. A search for economic evaluations of oral cancer screening was performed on Medline, CINAHL, Cochrane, PubMed, health technology assessment databases, and EBSCO Open Dissertations. The quality of studies was appraised using QHES and the Philips Checklist. Data abstraction was based on reported outcomes and study design characteristics. Of the 362 studies identified, 28 were evaluated for eligibility. The final six studies reviewed consisted of modeling approaches (*n* = 4), a randomized controlled trial (*n* = 1), and a retrospective observational study (*n* = 1). Screening initiatives were mostly shown to be cost-effective compared to non-screening. However, inter-study comparisons remained ambiguous due to large variations. The observational and randomized controlled trials provided considerably accurate evidence of implementation costs and outcomes. Modeling approaches, conversely, appeared more feasible for the estimation of long-term consequences and the exploration of strategy options. The current evidence of the cost-effectiveness of oral cancer screening remains heterogeneous and inadequate to support its institutionalization. Nevertheless, evaluations incorporating modeling methods may provide a practical and robust solution.

## 1. Introduction

Oral cancer has always been a significant disease burden worldwide. In 2020 alone, 377,713 new cases and 177,757 deaths were due to lip and oral cancer [1]. Its incidences also varied greatly based on geographical regions. This was partly due to differences in socio-demographic factors and lifestyles. For example, 65.8% of the global incidence and 74% of deaths occurred on the Asian continent [1]. The high incidences here were believed to be due to smoking and chewing of betel quids with areca nuts and tobacco [2]. To reduce this morbidity and mortality, numerous effective interventions and technologies have been recommended and introduced. However perplexingly, even with such improvements and advancements, the 5-year survival rate of oral cancer showed only minimal changes in the past decades [3,4,5].

One of the reasons postulated for the plateau and poorer outcomes was the delay in early detection, leading to diagnosis in later stages. Relative to other cancers, oral cancers are often preceded by visible and abnormal lesions, known as oral potentially malignant disorders (OPMD). They consist of a wide range of disorders such as oral leukoplakia, erythroplakia, submucous fibrosis, and lichen planus. These lesions have variable chances of progressing into neoplasms, ranging from as low as 5% to as high as 85% [6]. Although OPMD and oral cancer can be easily screened via visual inspection and palpation of the oral mucosa by trained personnel, they are frequently detected late. These then cascade to delays in early interventions and risk management, which impact future morbidities and mortalities [7,8,9].

Early screenings have thus been promoted as being an integral part of national control programs [10]. Studies on various oral cancer screening strategies have consistently demonstrated their feasibility, accuracy, and positive predictive values [7]. However, certain approaches, such as population-based screening, remained unendorsed given the lack of evidence on their cost-effectiveness [7,10]. Such a void continues to hamper policymakers from making informed decisions to shift from existing treatment-oriented modules to those focused on primary prevention and early detection. Additionally, the lack of studies also limits the ability of researchers to identify feasible and reproducible study designs to fill this knowledge gap.

Current recommendations often cite randomized controlled trials (RCT) as the gold standard in healthcare economic evaluations. However, adopting such a design in oral cancer screening can be challenging due to its pathophysiology and latency. Alternatively, modeling approaches have recently been advocated as a more feasible solution in many cancer screening programs, including oral cancer [7]. Several types of models have been proposed, each varying in terms of complexity and prognostic accuracy. Regardless of the preference, a lack of consensus on model structures and critical areas for consideration remains a challenge to constructing a pragmatic and credible protocol.

A systematic review is widely acknowledged as an essential approach to informing decision-makers and researchers about the best practices and policies. The methodological differences between the modeling approach and trial-based economic evaluations limit the comparability of recommendations while necessitating the evaluation of the methodologies separately. This includes the application of different quality assessment tools and the interpretation of findings. For these reasons, this systematic review aims to summarize the outcomes and compare the design of economic evaluations on oral cancer screening to guide future study development.

## 2. Materials and Methods

### 2.1. Source and Search Strategy

The initial search for this systematic review was conducted on four major databases: Medline, CINAHL, Cochrane, and PUBMED; two databases for health technology assessment: the Centre for Reviews and Dissemination of the University of York and the International Network of Agencies for Health Technology Assessment (INAHTA); and relevant theses in EBSCO Open Dissertations. Appropriate Medical Subject Headings (MeSH), thesauruses, and specific keywords were combined using Boolean operators in search terms and guided by the criteria in Table 1.

Examples of search terms are attached in Appendix A. The search was restricted to publications from 1 January 2000, until 31 December 2021. A more recent time frame was selected to ensure the outcomes evaluated were reflective of the present risk factors and treatment advancements. The bibliographical reference lists of selected studies for full-text screening were also hand-searched for other relevant studies. Papers were not confined to the English language. However, studies were only reported if a full copy in English was available for abstraction and quality assessment.

### 2.2. Study Selection

Titles and abstracts were first scrutinized by a reviewer (S.R.) based on relevance. The full text of selected studies was later obtained and reviewed independently by two reviewers (S.R. and B.Y.). There were no restrictions in terms of the population characteristics, patient recruitment methods, study design, or implementation of strategies to ensure a comprehensive review. Studies were only excluded if (1) the full text was unavailable, (2) there was no or a partial economic evaluation, (3) screening consisted of other than visual oral examinations, and (4) it was not an original article. At this stage, excluded studies were listed, and the reason for exclusion was reported. Any discrepancy in the selection of papers was discussed between the reviewers until a mutual agreement was reached.

### 2.3. Quality Assessments

Studies were first ranked by both reviewers according to quality scores using the broad Quality of Health Economic Studies (QHES) checklist. The list consists of 16 items of varying importance [11]. Additionally, a simplified Philips checklist was used to further evaluate studies adopting modeling methods, as per the recommendations from Cochrane and the National Institute for Health and Care Excellence (NICE) [12,13,14].

### 2.4. Data Extraction and Analysis

Data were extracted into three major sections to aid discussions: (1) study characteristics, (2) reported outcomes, and (3) model design. A standardized data extraction form was developed in Microsoft Excel based on literature and expert review [12]. The extracted information was summarized into evidence tables based on the set sections. The data were extracted by both reviewers (S.R. and B.Y.) and counter-referenced for completeness and accuracy.

Evaluations that piggybacked on top of clinical studies were labeled as trial-based economic evaluations. On the other hand, model-based evaluations consisted of any studies combining a variety of source data via an economic model [12]. This roughly ranges from decision trees, state-transition models, discrete event simulation models, and stochastic models. All models were evaluated for their construction and limitations. The outcomes examined included the effects, costs, resources used, incremental cost-effectiveness ratio (ICER), sensitivity analysis, and decision recommendations. Cost data are presented as per the reported year and currency.

## 3. Results

The initial search yielded 362 papers. After the removal of duplicates and irrelevant studies, the remaining 28 studies were assessed for eligibility. A total of six studies were included in the final review (see Figure 1). A detailed list of reasons for the excluded studies (*n* = 22) is available in Appendix A. All six studies were deemed high-quality based on the QHES checklist, with scores above 75 out of 100 (Appendix A). Inadequate length of the analytic horizon and lack of discounting for benefits and outcomes were the most frequent criteria (*n* = 3) not fulfilled. The reduced quality score for Van der Meij et al. [15] was partially due to the inability to obtain the referenced study model. This impacted the evaluation of criteria such as model perspectives in addition to the methodology for cost estimates not being characterized.

### 3.1. Study Type and Population

Study design, screening strategies, and outcome measures are summarized in Table 2. They varied in the design of their evaluations: modeling methods (*n* = 4), randomized controlled trial (*n* = 1), and retrospective observational study (*n* = 1). The studies predominantly explored the impact of the initiation of oral cancer screening in populations aged between 35 and 40 years. While Van der Meij, et al. [15] began the decision model using the Netherlands population above the age of 15, the oral lichen planus prevalence was still presumed to occur after the age of 55. Likewise, Huang et al. [16] reported the average age of OPMD diagnosis in the Taiwan population as 55.4 ± 12.5 years. These values were consistent with the reported mean age of the population at risk of developing oral cancer and OPMD in the literature [17,18].

### 3.2. Screening Strategies

The strategies explored mostly consisted of population-based screening (*n* = 4), followed by targetted screening (*n* = 2), and opportunistic screening (*n* = 1). They were all compared with ‘no screening’ in their respective evaluations. The Health Promotion Administration in Taiwan integrated its screening program into public healthcare via the provision of free biennial oral mucosal examinations for high-risk citizens [16]. They consisted of adults with smoking or betel quid chewing habits in addition to aboriginals over the age of 18 years. Similarly, Dedhia et al. [21] modeled annual community screening of high-risk males with regular use of tobacco and alcohol. Other studies adopted a broader approach, screening everyone regardless of their risk factors [15,19,22]. Through simulation of different possible strategies, Speight et al. [20] explored the modality of having an invitational and opportunistic screening for all patients in general medical and dental practice, in addition to those at high risk.

### 3.3. Cost Perspectives

Only three studies opted for a societal perspective in their cost analysis. Subramanian et al. [19] had the most detailed programmatic cost components, consisting of training, recruitment, the screening process, administrative work, and the provision of educational messages. The total societal cost included research, diagnostics, treatment, and loss of patient productivity. Kumdee et al. [22] also incorporated extensive direct medical and non-medical costs. Even though Dedhia et al. [21] took a similar societal perspective in their evaluation, the total cost did not include patient-related expenditures. Furthermore, program development and management costs were deliberately not included. The rest of the studies adopted a healthcare provider’s perspective.

### 3.4. Screening Outcomes

Most of the outcomes of the studies were generated following a yearly cycle and a lifetime horizon. Such an approach allowed the impact of screening initiatives on the variable malignant transformation rates (MTR) of OPMD to be established over the life span of patients. Van der Meij et al. [15] estimated the long-term outcomes by calculating the equivalent lives saved according to a 25-year life expectancy and changes in quality-adjusted life years (QALYs). Similarly, Huang et al. [16] extrapolated the life expectancies and projected years of life lost by calculating the survival functions of their oral cancer cohort and the general population.

On the contrary, the RCT in India focused on the real-time outcomes from three screening cycles over a shorter period of nine years [19]. They measured outcomes in terms of the number of cancer cases detected, cancer deaths, and the corresponding life-years saved, based on the assumption that cancer death occurs at the age of 50. The rest of the three modeling studies evaluated the QALYs gained through a projected lifetime implementation of the screening programs [20,21,22].

It is worth highlighting that three out of the four modeling studies had similar estimates for health utility values [15,20,21]. The values were obtained from a study using a standard gambling questionnaire among employees of a commercial company in England [23]. The study identified three health states for the utilities: precancer, early (Stage I), and late cancer (Stages II to IV). Dedhia et al. [21] used the early cancer utility value to represent Stages I and II, while the rest applied similar categorization as the original study. Only Kumdee et al. [22] obtained a current and local utility value through patient interviews.

### 3.5. Incremental Cost-effectiveness Ratio

The ICER of strategies is summarized in Table 3. The values demonstrated a large disparity, reflecting the wide heterogeneity arising from the study’s conduct and setting. For example, community screening by healthcare workers in India reported the lowest ICER value at USD 835 per life-year saved. The study by Kumdee et al. [22] also reported relatively lower ranges of ICER values at THB 82,292 to 311,030 (USD 2377 to 8984) per QALY, albeit forming almost double the country-specific threshold at the higher end. On the other hand, the values from developed countries were relatively higher. The simulation of population screening in the Netherlands, for instance, reported an ICER of USD 53,430 per equivalent life saved, while the various scenarios explored by Speight et al. [20] reported ICER values ranging from GBP 13,285 to GBP 48,468 (USD 24,444–89,181) per QALY.

The threshold adopted appeared to be dependent on the availability of a national willingness-to-pay threshold and local recommendations for conducting an economic evaluation. Both Huang et al. [16] and Subramanian et al. [19] established the cost-effectiveness of the screening strategies based on one gross domestic product (GDP) per capita. In contrast, Dedhia et al. [21] and Kumdee et al. [22] applied a nationally accepted willingness-to-pay threshold, while Speight et al. [20] used a benchmark value range for a QALY gained to determine their cost-effectiveness. These values ranged extensively from USD 2900 in India to USD 75,000 in the USA [19,21]. No threshold was provided by Van der Meij et al. [15], thus limiting the ability to discern the cost-effectiveness of the screening strategies.

### 3.6. Modeling Approaches to Economic Evaluation

Three studies applied state-transition Markov modeling to the economic evaluation of the screening programs [20,21,22], while Van der Meij et al. [15] adopted a simpler decision model approach. To model the natural history, the severity of tumors was staged following the TNM system by the American Joint Committee on Cancer, ranging from Stage I to Stage IV. Two of the Markov models adopted these stage-specific discrete health states based on severity. Dedhia et al. [21] simplified the health states by combining the stages of early cancer (Stage I/II) and late cancer (Stage III/IV), which correspond to the significant differences in the management, survival, and mortality risks. Van der Meij et al. [15] in contrast, categorized the health states into healthy, Stage I, and ‘Stage II and above’.

All the models adopted a stage-shift approach to the screening effect, with varying methods. In the decision model by Van der Meij et al. [15], this was emulated by an external shift, where the screening program was assumed to lead to a larger proportion of diagnoses made at an earlier stage (Stage I) relative to a ‘no screening’ approach. The rest of the Markov models replicated this through the incorporation of higher compliance with screening and/or biopsy, generating a larger proportion of patients being screened and treated earlier [20,21,22]. Furthermore, the studies also simulated better outcomes by detecting individuals in the same stages but earlier. This internal shift was mimicked indirectly by incorporating modifiers such as reducing the MTR, clinical upstaging, or probabilities of death, in addition to increasing the rate of regression of precancers. These were possible as the Markov models incorporated dual-prognostic models: undetected patients and patients that are screened and diagnosed for treatments.

### 3.7. Quality and Validity of Models

The four modeling studies were further assessed for risk of bias using the simplified Philips checklist (Appendix A). All the state transition modeling studies were generally well executed, and efforts were made to ensure the quality of the recommendations. Both Speight et al. [20] and Kumdee et al. [22] managed to tackle most of the relevant components, such as specifying the decision-maker, evaluating all feasible options, and having model inputs that were consistent with their perspectives. It is worth noting that only Speight et al. [20] dealt with heterogeneity in the model by running it separately for different gender and age groups. The most common criteria not fulfilled overall were conducting half-cycle corrections to both cost and outcomes and a lack of justifications for their omissions.

Both Van der Meij et al. [15] and Dedhia et al. [21] did not specify the primary decision maker and excluded the cost of screening programs, which was deemed partially unjustified as they aimed to explore programmatic implementations. Lastly, similar to the evaluation using the QHES checklist, the study by Dedhia et al. [21] scored lower due to a lack of detail in the model’s perspective and structure, time horizon, and discounting.

Kumdee et al. [22] carried out both face validation and internal validation by extrapolating the predicted survival from local observation data. Their predicted survival curve demonstrated that the five-year survival rate of the baseline model was close to the local observational study. Similarly, Dedhia et al. [21] tested the validity of their model mid-cycle with reported precancer prevalence and annual cancer death data. The model values were shown to coincide with findings in the United States, although the death rate was at the higher end of the reference range.

### 3.8. Sensitivity Analysis

It is worth highlighting that while sensitivity analysis was not possible for both the observational study and the RCT, the studies did point out the potential sample uncertainties arising from sampling and data organization. All the modeling studies carried out various sensitive analyses for a range of inputs. Dedhia et al. [21] found, for example, that parameters such as MTR, willingness-to-pay threshold, compliances, and the probability of participating in an oral examination affected the outcomes of oral cancer screening programs in a one-way sensitivity analysis. Analysis by Kumdee et al. [22] similarly demonstrated that the ICER value was sensitive to the specificity and sensitivity of self-mouth examination and visual examination by a trained dentist, in addition to compliance with the screen.

Both Speight et al. [20] and Kumdee et al. [22] further conducted a probabilistic sensitivity analysis to illustrate the uncertainties of the input parameters. Kumdee et al. [22] predicted that, based on the inherent uncertainties, the base-case screening program has only a 30% chance of being cost-effective. The ICER values reported by Speight et al. [20] varied by age and sex while being most labile to the treatment effect of OPMD. In their model, a higher treatment effect of 10 and 20 % reduction in the MTR of OPMD made the opportunistic screening programs more favorable to people 40 to 70 years old relative to those over the age of 70. The uncertainties associated with oral cancer screening were further evidenced to generate an expected value of perfect information (EVPI) from GBP 8 to 462 million (USD 14.7 to 850 million). The EVPIs were high for all the parameters and varied according to subgroups, with MTR, disease progression, self-referral rates, and treatment costs generating the highest value for uncertainties.

## 4. Discussion

Overall, the total number of studies on screening programs for oral cancer was rather small compared to other types of cancer, such as prostate and breast cancer [24,25]. Furthermore, from 2021 to 2023, no new studies were reported based on a recent review of oral cancer screening approaches and a literature summary [26,27]. This reflected the fact that although many oral cancer screening programs were recommended and instituted at various levels, they were often not evaluated rigorously in terms of their economic benefits. It was also evidenced that there is no consistent and established study design for the economic evaluation of oral cancer screening.

Nevertheless, newer studies incorporating modeling approaches, even though limited, are showing potential for application in future research to fill these evidence gaps (Table 4). They are seen as favorable, specifically in oral cancer and OPMD, as the MTR varies widely and the disease progresses over a long period. Additionally, observational studies or analyses piggybacked on controlled trials may not be able to capture the clinical upstaging, variations in risks, annual treatment costs, and quality of life if they are not carried out in a real-life period. Modeling approaches provide a feasible alternative to capture these long-term consequences with minimal use of resources. On the flip side, these studies are dependent on the quality of the evidence in terms of the comprehensiveness of the costs and outcomes associated with the actual implementation of the program and the process involved. Our review evidenced that many of the cost components, such as the human resources needed, are often not substantially included in the modeling approaches.

### 4.1. Cost-Effectiveness of Screening Strategies

Generally, all studies reported that several implementations of screening initiatives were cost-effective compared to not screening for oral cancer. It was, however, difficult to provide recommendations on the best screening options as the evidence differed based on country, setting, payer system, costing approach, and parameters modeled. Nevertheless, a valid comparison between screening measures can still be extracted from the various scenarios modeled by Speight et al. [20] and Kumdee et al. [22]. The exploration of different implementation options based on a similar setting allowed the implications of such strategies to be gauged.

In the setting of a developed country (the United Kingdom), population-based screening was shown to incur higher costs compared to opportunistic screening. This was predominantly contributed by the huge amount of resources required to cover a larger population. Nevertheless, if the coverage was narrowed to either opportunistic screenings or focused on high-risk individuals, the screening strategies showed improvements in terms of their ICER values [20]. In short, the accessibility and recruitment of individuals for screening are important variables that impact the cost-effectiveness of the strategies.

The study in Thailand, on the other hand, gave an alternative paradigm of a developing Asian country. They evidenced that strategies that involved multiple healthcare personnel incurred a higher cost but with minimal improvements in terms of QALY gained. Thus, it was critical for a balance to be achieved between the accuracy of the diagnosis and the number of personnel involved in the screening process. Encouraging oral self-examination facilitated a reduction in dentists’ workload and made the program more feasible on a national scale. Additionally, in situations where there are insufficient dentists in the public healthcare system, a well-trained dental nurse can be considered for the success of these screening programs [22].

### 4.2. Guidelines for Future Models

At the current moment, the natural history model for oral cancer with its wide array of premalignant lesions and conditions remains to be validated [18]. An investigation of the epidemiological and physiological progress of OPMD alongside modeling will be able to further add predictive value to the study. However, the incorporation of multiple intrinsic factors, such as types of OPMDs and risk factors, may inadvertently complicate the structure and reduce their intuitiveness for decision-making [28]. For broader national-level policies, the inconsistency in the MTR of various conditions may need to be traded off for the applicability of decisions.

Health states should be based on the TNM staging system. This is because it allows the states to be informed about management decisions according to practice and treatment guidelines. All three of the Markov state transition models reviewed here establish that such a design can mirror real-world outcomes and envisage future costs. Models must also incorporate both treated and untreated cohorts to reflect the reality of the current population mix. Despite the predictability of such cure models, which are yet to be established, it avoids the overestimation of comparisons made solely with the untreated group.

Due to the scarcity of published studies, the intuitiveness of the Markov models could not be contrasted with other methods such as discrete-event simulation and stochastic models. Based on this review, the cohort-level simulation was less intricate, easily simulated, and needed less computational time for analysis while maintaining its robustness. The key drawback of the Markovian process being ‘memoryless’ was also demonstrated to be easily countered by incorporating tunnel states into the existing model structure [20,21]. This is crucial, as probabilities such as precancer tissue regression and cancer reoccurrence are time-dependent. Assimilation of tunnel states avoids the overvaluation of treatment benefits and MTR.

All four of the modeling studies incorporated QALY to capture the morbidity and mortality of oral cancer progression and treatment effects. While they are a good proxy to corroborate the effectiveness of screening programs, utility values need to be accurate and reflective of the investigated population and time frame. The QOL values based on the findings in England in 1997 in three of the studies, for instance, may not represent the current local healthcare system and patient responses [23]. Furthermore, the adopted values from a healthy population may overestimate or minimize disease impact [29]. As the debate on the patient-versus-general population continues, investigators need to consider that population values might not accurately reflect the true benefits achieved by patients from the screening programs [30]. Using disease-specific QOL tools and conducting sensitivity analyses on utility values may provide policymakers with better evidence.

It is also imperative for researchers to adopt modeling approaches to conduct extensive sensitivity analysis and explore strategy-specific challenges. This is because the assumptions and input parameters can alter the direction of the cost-effectiveness of strategies considerably. Parameters such as MTR, screen and treatment compliances, screening performances, treatment effects, and incidences should primarily be explored for all possible ranges in the population. Although the deterministic sensitivity analysis was sufficient to characterize these uncertainties, a probabilistic sensitivity analysis may better establish the structural uncertainties of the model [31]. Additionally, it is recommended future simulations put forth sufficient effort to ensure more accurate local values and variability by demography are obtained.

### 4.3. Strengths and Limitations

The most significant strength of this review is its extensive, in-depth analysis of study designs for oral cancer screening evaluations. The comparison of designs and conduct allows future investigators to adopt feasible and robust methods for their setting. Additionally, the study reiterates the gaps in the reliability of the evidence for the cost-effectiveness of screening strategies, which prevent their recommendation and institutionalization. Areas such as the inclusion of comprehensive costing components and the incorporation of more recent and local health utility values are critical for steering decisions on the implementation of screenings. With a huge population scope, high programmatic costs, and numerous variations in conduct and values, the review demonstrates the modeling approach as a feasible solution to fill the knowledge gaps before initiating more resource-intensive trials.

One of the major limitations of the review is that the disparity of studies does not allow for a direct comparison of cost-effectiveness values and study quality. This was best illustrated in terms of the cost implications, where the broader societal perspective led to larger financial implications compared to those from the perspectives of healthcare providers alone. The lack of patient expenditures and indirect costs underestimates the economic consequences and limits the interpretability of findings in different settings.

In our review, a compromise was made to adopt more lenient inclusion criteria to represent and discuss numerous study designs instead of focusing on inter-study outcomes. This aimed to encourage the construction of study methods to tackle the lack of economic evaluation in oral cancer screening. As echoed by the Cochrane Oral Health Group and US Preventative Services Task Force (USPSTF), the cost-effectiveness of visual examinations in oral cancer screening can only be evaluated and established when a wealth of information has been accrued [7,32].

This review also did not include gray literature or studies in other languages. Furthermore, while numerous oral screening programs are conducted around the world as part of national healthcare services or institution-driven, these programs are often neither reported nor evaluated economically. This could significantly skew the review’s narrative and focus. Appraising, reporting, and compiling these initiatives can help build up the sufficient evidence needed to build local models and assist in decision-making at a national level.

## 5. Conclusions

The current evidence of the cost-effectiveness of oral cancer screening is still heterogeneous and inadequate to support its institutionalization. However, the available data points toward certain implementational strategies being possibly cost-effective. While trials and observational studies were able to provide accurate information on the cost of programs, modeling approaches appeared favorable as they could generate a wealth of information for decision-making. The state-transition model with built-in tunnel states was able to reflect the impact of screening strategies adequately. As the models and decisions were shown to be very sensitive to assumptions and input parameters, the review reinforced the need for more critical sensitivity analysis and validation steps. We hope that our findings can encourage a more robust exploration of economic evaluations in oral cancer screening to fill the current knowledge gaps.

## Figures and Tables

**Figure 1 healthcare-11-01198-f001:**
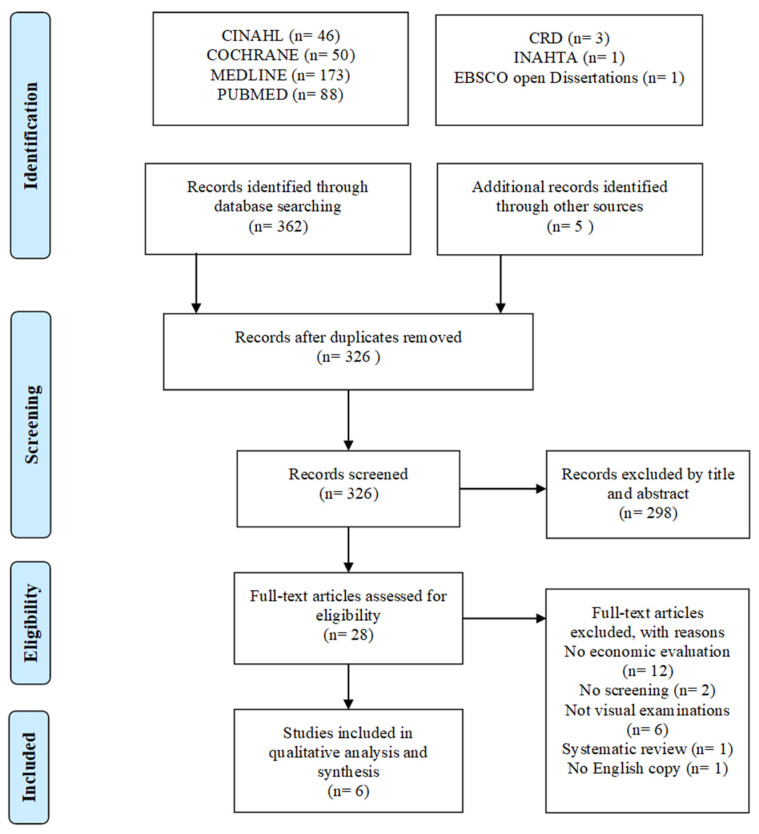
Flow chart of literature search.

**Table 1 healthcare-11-01198-t001:** Search concepts and examples of phrases used.

Concept	Examples of Similar Phrases
OPMD	Oral leukoplakia, pre-malignant, and oral lesion/dysplasia.
Oral cancer	Oral cancer/neoplasm/carcinoma, and squamous cell carcinoma.
Screening	Screening, diagnosis, oral examination, and early detection.
Economic evaluation	Economic evaluations, cost-effectiveness, cost-benefit, and cost-utility.

**Table 2 healthcare-11-01198-t002:** Summary of studies included for quality assessment and analysis.

Authors (Year), Country	Study Design	Perspective	Population	Time Horizon	Strategies (Visual Examination)	Outcomes Measured
Subramaniam, et al. (2009) [19], India	Randomized controlled trial	Societal	Population in 13 municipal units, over 35 years	9 years	No screening; 3-yearly screening (THW)	Cancers detected, life-years saved, costs, ICER
Huang et al. (2019) [16], Taiwan	Retrospective observational study	Healthcare system	All populations between 2010 and 2013	Lifetime	No screening; Biannual, free (MP) ^3^	Life expectancy, EYLL, lifetime medical cost, and ICER.
Meij et al. (2002) [15], Netherland	Decision-analytic model	Healthcare system ^1^	Hypothetical population with OLP over 55 years old	25 years	No screening (standard care and educational messages); Population screening by OS; Population screening (DP)	QALYs, equivalent lives saved, and costs.
Speigh et al. (2006) [20], UK	Markov state-transition model	Healthcare system	Hypothetical healthy population over 40 years old	60 years	No screening; Invitational screening (MP); Invitational screening (DP); Opportunistic screening (MP); Opportunistic screening (DP); Opportunistic ‘high-risk’ screening (MP); Opportunistic ‘high-risk’ screening (DP); Invitational screening (OS)	QALYs, lifetime costs, and ICER.
Dedhia et al. (2011) [21], USA	Markov state-transition model	Societal	Hypothetical population of high-risk males ^2^ over 40 years old	40 years	No screening; Annual, community-based screening by trained health workers	Annual cancer deaths, QALYs, costs, and ICER.
Kumdee et al. (2018) [22], Thailand	Markov state-transition model	Societal	Hypothetical healthy population over 40 years old	60 years	No screening; MSE + screening by TDN, DP, and OS; Screening by TDN, DP, and OS; Screening by DP and OS; MSE + screening by DP and OS	QALYs, lifetime costs, and ICER.

OLP—oral lichen planus; MP—general medical practitioner; DP—general dental practitioner; MSE—self mouth examination; TDN—trained dental nurse; OS—oral surgeon; EYLL—expected years of life lost. ^1^ not explicitly stated in the paper and conclusion obtained from discussion among panel; ^2^ high-risk defined as age over 40 years with recent, regular use of tobacco and/or alcohol; ^3^ free for Taiwan residents with smoking and/or betel quid chewing habits who are at least 30 years old or at least 18 years old (if they are aboriginal).

**Table 3 healthcare-11-01198-t003:** Summary of ICER values by screening strategies.

Study (Country)	Strategy	Screener	ICER	Threshold
Subramaniam et al. [19] (India)	3-yearly screening in municipal units	THW	USD 835/LYS (all) USD 156/LYS (high risk)	USD 2900
Huang et al. [16] (Taiwan)	Biannual, national, screening program	MP	USD 28,516/LYS (all) USD 2515/LYS (Stage 0) ^1,2^ USD 5579/LYS (Stage 1) ^1^	USD 25,873 ^3^
Meij et al. [15] (Netherland)	OLP population screening	OS	USD 2137/QALY USD 53,430/LYS	USD 53,430 per LYS
		DP	USD 1339/QALY	
Speigh et al. [20] (UK)	Invitational population screening	MP	GBP 26,586/QALY	GBP 20,000–30,000 per QALY
		DP	GBP 28,160/QALY	
		OS	GBP 39,300/QALY	
	Opportunistic population screening	MP	GBP 24,149/QALY	
		DP	GBP 23,367/QALY	
	Opportunistic ‘high-risk’ screening	MP	GBP 23,118/QALY	
		DP	GBP 23,147/QALY	
Dedhia et al. [21] (USA)	Community-based ‘high-risk’ screening	THW	−USD 6232/QALY ^2^	USD 75,000 per QALY
Kumdee et al. [22] (Thailand)	Population screening	MSE + TDN + DP + OS	THB 320,618/QALY	THB 160,000 per QALY
		TDN + DP + OS	THB 174,621/QALY	
		DP + OS	THB 100,016/QALY	
		MSE + DP + OS	THB 82,292/QALY	

THW—trained health workers; LYS—life-year saved; OLP—oral lichen planus; MP—general medical practitioner; DP—general dental practitioner; OS—oral surgeon/specialist; TDN—trained dental nurse; MSE—mouth self-examination. ^1^ outcomes if the cases were followed intensively and diagnosed at stage 0 or 1; ^2^ cost-saving (dominant); ^3^ estimated from literature based on reported details.

**Table 4 healthcare-11-01198-t004:** Strengths and weaknesses of study designs for economic evaluation based on reviewed literature.

Design	Strengths	Weaknesses	Considerations
Randomised controlled trial	Allows for an accurate and complete estimation of programmatic costs incurred. Effectiveness is measured in real-time. A valid control prevents overestimation or underestimation of outcomes. Able to obtain demographic and clinical information for management. Accurate information on inputs such as MTR, incidence, and detection rate.	Needs a high capacity of human and financial resources. Effectiveness is limited to the study period. Long-term consequences and costs, such as the extension of life expectancy and productivity, are not able to be captured. It is cost-intensive to explore multiple possible strategies. Findings might be limited to the population investigated, and extrapolation is still needed for a wider national policy.	Ensure the availability of sufficient resources and support. The screening strategy should be well investigated and feasible for implementation. Ensure the length of the study period is comparable to effectiveness outcomes, such as 5-year survival. A diverse population and subgroup analysis may assist in identifying specific targets for screening. Additional projections of outcomes and costs over the total life span of the population (with appropriate discounting).
Retrospective observational study	Allows for the precise estimation of clinical outcomes. Controlled trials are easier to conduct and consume fewer resources compared to uncontrolled trials. Effectiveness is measured in real-time. Able to obtain demographic and clinical information for sub-group analysis. Accurate patient-specific direct medical cost.	Highly dependent on the availability and quality of data. Needs an extended period of observation to establish long-term consequences. Effectiveness is limited to the study period. Long-term consequences and costs, such as the extension of life expectancy and productivity, cannot be captured. Unable to obtain societal or indirect medical costs. Programmatic costs may be underestimated if they cannot be distinguished from medical records.	A well-established and interlinked registry, national databases, and clinical records. Availability of medical cost/expenditure data, reimbursement data, or universal coverage. Additional estimation and discounting of outcomes and costs over the total life span of the population (if the observation period is short). Subgroup analysis (comorbidities or risk factors) may assist in identifying specific targets for screening.
Decision analysis	Fastest and least resource-consuming. Outcomes can be simulated easily for a range of variables or strategies. Crude long-term outcomes can also be estimated via the incorporation of sufficient parameters and assumptions.	Requires a good and validated decision structure to simulate the disease and the screening progress. Outcomes are highly dependent on the quality of the information and the intuitiveness of the decision structure.	The model needs to be able to reflect the long-term outcomes well and be validated. Sufficient efforts should be focused to ensure the accuracy and robustness of the information applied. Capitalize on the approach by exploring various implementational strategies. Conduct an extensive sensitivity analysis.

## Data Availability

The data presented in this study are openly available from the Harvard Dataverse at [https://doi.org/10.7910/DVN/M3HODO], accessed on 7 January 2023.

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
