# Peer review of "Economic Evaluation of Oral Cancer Screening Programs: Review of Outcomes and Study Designs"

_healthcare, 2023, doi:10.3390/healthcare11081198_

Round 1
Reviewer 1 Report
The review is easy to read and well-organized.
However, the introduction section needs to discuss the need for a systematic review of economic evaluation.
Three of the included studies took a societal perspective. I suggest the authors discuss the limitation of not including all cost categories for the specific perspective.
The authors discussed the sensitivity analysis for model-based economic evaluation (input parameter uncertainty). I suggest the authors discuss sample uncertainty for the Randomized clinical trial and observational studies.
Author Response
The review is easy to read and well-organized.
However, the introduction section needs to discuss the need for a systematic review of economic evaluation.
- We thank the reviewer for the kind words and suggestions. We have included a paragraph in the Introduction section [page 2, line 76-83] to detail the need for a systematic review of the economic evaluations.
Three of the included studies took a societal perspective. I suggest the authors discuss the limitation of not including all cost categories for the specific perspective.
- We agree with the author and had added the example of the cost perspectives to illustrate the limitations of not including other cost perspectives [page 11, line 415-420]
The authors discussed the sensitivity analysis for model-based economic evaluation (input parameter uncertainty). I suggest the authors discuss sample uncertainty for the Randomized clinical trial and observational studies.
- We acknowledge this gap and had added a short explanation of the issue [page 8, line 286-288].
- We were not able to expound the point further as we aimed to focus the discussions on the modeling approach.
Reviewer 2 Report
Overall, the article has the potential to make an important contribution to the literature on economic evaluations of oral cancer screening programs. By incorporating these suggestions, the manuscript can be improved and better serve the needs of its readers,but a few comments to improve the article.
Discuss the implications of the findings: The article should discuss the implications of the findings for policymakers and healthcare providers. This could include a discussion of the cost-effectiveness of oral cancer screening programs and the potential benefits and drawbacks of different screening strategies.
The authors should incorporate findings from screening of other types of cancers and use them to shed light on how to improve oral cancer screening.
Improve the clarity and organization of the article: The article would benefit from improved clarity and organization. This could include breaking up long paragraphs, using subheadings to organize the content, and ensuring that the language is accessible to a wide audience.
Author Response
Overall, the article has the potential to make an important contribution to the literature on economic evaluations of oral cancer screening programs. By incorporating these suggestions, the manuscript can be improved and better serve the needs of its readers,but a few comments to improve the article.
Discuss the implications of the findings: The article should discuss the implications of the findings for policymakers and healthcare providers. This could include a discussion of the cost-effectiveness of oral cancer screening programs and the potential benefits and drawbacks of different screening strategies.
- We appreciate the feedback and we agree with the reviewer. We have added a subheading 4.1 under the Discussion to briefly elaborate on the potential implications of the various screening options [page 9, line 329-353].
The authors should incorporate findings from screening of other types of cancers and use them to shed light on how to improve oral cancer screening.
- We greatly appreciate this suggestion. While we hoped to expound on this suggestion, this study aimed to be focused on the design of the economic evaluations to explore the oral cancer screening, instead of the conduct of the screenings themselves. This was incorporated in the discussion section [page 8, line 309-316]
- We have also discussed about some strategies that can be adopted under Discussion subheading 4.1 [page 9, line 329-353]
Improve the clarity and organization of the article: The article would benefit from improved clarity and organization. This could include breaking up long paragraphs, using subheadings to organize the content, and ensuring that the language is accessible to a wide audience.
- Thank you for the feedback. We have relooked into some of the paragraphs and added some further explanations to improve the clarity of our points.
- For example, we have added some justifications in terms of the rationale for the review and provided additional explanations for the sensitivity analysis.
- We have also broken some of the larger sections into shorter content based on the idea we would like to share with the readers.
- Subheading 3.7 was broken further into subheadings 3.7 Quality and Validity of Models and 3.8 Sensitivity Analysis.
Reviewer 3 Report
The article " Economic Evaluation of Oral Cancer Screening Programmes: Review of Outcomes and Study Designs" by Sivaraj Raman, Asrul Akmal Shafie, Bee Ying Tan and Sok Ching Cheong, is a systematic review, which compares cost-effectiveness over economic evaluations for oral cancer screening programs. An electronic search was performed in the PubMed database, Medline, CINAHL and, Cochrane. It is a very well planned, conducted, and written study. It is an interesting and relevant article, and the structure of the manuscript seems adequate.
The authors present a well-edited and scientifically consistent systematic article in a very relevant area in dentistry. Overall, the manuscript left me with a good impression, and it is an important review in the field of public health in dentistry.
The methodology is well described with sufficient data and results to support the review. The conclusion is in accordance with the objective outlined by the authors and is supported by the results obtained. Conclusions are clear. Bibliographic references are in accordance with the journal's norms.
Author Response
The article " Economic Evaluation of Oral Cancer Screening Programmes: Review of Outcomes and Study Designs" by Sivaraj Raman, Asrul Akmal Shafie, Bee Ying Tan and Sok Ching Cheong, is a systematic review, which compares cost-effectiveness over economic evaluations for oral cancer screening programs. An electronic search was performed in the PubMed database, Medline, CINAHL and, Cochrane. It is a very well planned, conducted, and written study. It is an interesting and relevant article, and the structure of the manuscript seems adequate.
The authors present a well-edited and scientifically consistent systematic article in a very relevant area in dentistry. Overall, the manuscript left me with a good impression, and it is an important review in the field of public health in dentistry.
The methodology is well described with sufficient data and results to support the review. The conclusion is in accordance with the objective outlined by the authors and is supported by the results obtained. Conclusions are clear. Bibliographic references are in accordance with the journal's norms.
- We greatly appreciate the feedback and kind words from the reviewer. We truly echo the reviewer’s opinion in that the study is important for us to be able to provide some guidance in oral cancer screening, from a public health perspective.
Reviewer 4 Report
Abstract:
The abstract is well done, it describes the content of the study very well and allows the potential reader to decide whether the topic of the manuscript is of interest to him/her or not. I think that the current form of the abstract is good enough for a possible publication.
The 6 keywords are well chosen, sufficient and suggestive for the manuscript.
On a scale of 1 to 10, I’ll give 9 points for the abstract.
Introduction:
The authors manage, in approximately one page of text, helped by 10 bibliographic references, to convince us of the necessity and usefulness of the study they have carried out. The level of introduction is good, even if it does not impress.
On a scale of 1 to 10, I agree 8 points for introduction.
Methodology:
This is a consistent, good chapter, which describes in general terms how the literature review was done, step by step. It does not give reproducibility to the study, but it leaves the impression of a properly conducted review. Also, they do not specify clearly what type of literature review they proposed, the reader can only guess a systematic review, because there are not enough elements for a meta-analysis.
On a scale of 1 to 10, I agree 9 points for methodology.
Results:
This is the most consistent chapter of the manuscript; it is very well done, also supported from a graphic point of view. I appreciate the way in which the presentation of the results is thought out, the 7 subchapters follow a logical thread and naturally lead to the part reserved for discussions.
I don't find any deficiencies in this chapter and I think it deserves maximum marks.
On a scale of 1 to 10, I agree 10 points for results.
Discussion:
This is a correctly written chapter, but without impressing. However, I like that the authors try to draw "Guidelines for Future Models" and that they have the honesty to emphasize the "Strength and Limitations" of their study. This approach adds value to the manuscript and a grade close to the maximum.
In this situation, on a scale of 1 to 10, I agree 9 points for discussion.
Conclusion:
The conclusions are correctly written, respecting the general level of the manuscript. The authors do not forget to mention, even if very briefly, possible future research directions in this field.
On a scale of 1 to 10, I agree 8 points for conclusions.
Bibliography/References:
32 references, current, correctly written and correctly quoted in the text represent a reasonable level for an article; but, for a literature review, it is not enough to convince us of the depth of database research. I did not notice any citation errors.
On a scale of 1 to 10, I agree 7 points for the bibliography.
Figures/Tables:
I identified 4 tables and 1 figure, of good quality, necessary and useful for the manuscript. I think that for this literature review, the graphic part could be supplemented.
On a scale of 1 to 10, I agree 7 points for this chapter.
Review Decision:
Accept after minor revision.
Author Response
The abstract is well done, it describes the content of the study very well and allows the potential reader to decide whether the topic of the manuscript is of interest to him/her or not. I think that the current form of the abstract is good enough for a possible publication.
The 6 keywords are well chosen, sufficient and suggestive for the manuscript.
On a scale of 1 to 10, I’ll give 9 points for the abstract.
- We truly appreciate the positive feedback from the reviewer.
Introduction:
The authors manage, in approximately one page of text, helped by 10 bibliographic references, to convince us of the necessity and usefulness of the study they have carried out. The level of introduction is good, even if it does not impress.
On a scale of 1 to 10, I agree 8 points for introduction.
- Thank you for the feedback. We have also added a short section in the Introduction to explain the rationale for the systematic review [page 2, line 76-83]
Methodology:
This is a consistent, good chapter, which describes in general terms how the literature review was done, step by step. It does not give reproducibility to the study, but it leaves the impression of a properly conducted review. Also, they do not specify clearly what type of literature review they proposed, the reader can only guess a systematic review, because there are not enough elements for a meta-analysis.
On a scale of 1 to 10, I agree 9 points for methodology.
- Thank you for highlighting the issue. We have added the ‘systematic review’ in the abstract [page 1, line 22], the study objective [page 1, line 81] and the Methodology section [page 2, line 86]
Results:
This is the most consistent chapter of the manuscript; it is very well done, also supported from a graphic point of view. I appreciate the way in which the presentation of the results is thought out, the 7 subchapters follow a logical thread and naturally lead to the part reserved for discussions.
I don't find any deficiencies in this chapter and I think it deserves maximum marks.
On a scale of 1 to 10, I agree 10 points for results.
- Thank you for recognizing the team’s analysis and the kind words.
Discussion:
This is a correctly written chapter, but without impressing. However, I like that the authors try to draw "Guidelines for Future Models" and that they have the honesty to emphasize the "Strength and Limitations" of their study. This approach adds value to the manuscript and a grade close to the maximum.
In this situation, on a scale of 1 to 10, I agree 9 points for discussion.
- We truly appreciate the feedback.
Conclusion:
The conclusions are correctly written, respecting the general level of the manuscript. The authors do not forget to mention, even if very briefly, possible future research directions in this field.
On a scale of 1 to 10, I agree 8 points for conclusions.
- Thank you for the suggestion. We have added a line on the future research direction in the Conclusion [page 12, line 444-446]
Bibliography/References:
32 references, current, correctly written and correctly quoted in the text represent a reasonable level for an article; but, for a literature review, it is not enough to convince us of the depth of database research. I did not notice any citation errors.
On a scale of 1 to 10, I agree 7 points for the bibliography.
- We appreciate the feedback. We have conducted an extensive and systematic search, which yielded an initial 362 studies. However, as the question was narrowed to provide specific recommendations in terms of study design, the final references were reduced.
Figures/Tables:
I identified 4 tables and 1 figure, of good quality, necessary and useful for the manuscript. I think that for this literature review, the graphic part could be supplemented.
On a scale of 1 to 10, I agree 7 points for this chapter.
- Thank you for the feedback. We aimed to keep the figures concise to allow for more detailed elaboration.
Review Decision:
Accept after minor revision.
- We greatly appreciate the reviewer's suggestions and words of encouragement. We hope that we managed to further improve the quality of our article based on the constructive feedback.